# Impact of Maternal Mediterranean-Type Diet Adherence on Microbiota Composition and Epigenetic Programming of Offspring

**DOI:** 10.3390/nu16010047

**Published:** 2023-12-22

**Authors:** Tamlyn Sasaki, Megan Kawamura, Chirstyn Okuno, Kayleen Lau, Jonathan Riel, Men-Jean Lee, Corrie Miller

**Affiliations:** 1John A. Burns School of Medicine, University of Hawaii, Honolulu, HI 96813, USA; 2Department of Obstetrics, Gynecology and Women’s Health, John A. Burns School of Medicine, University of Hawaii, Honolulu, HI 96826, USA

**Keywords:** epigenetics, Mediterranean diet, maternal diet, meconium, cord blood, DNA methylation, maternal lifestyle, microbiome

## Abstract

Understanding how maternal diet affects in utero neonatal gut microbiota and epigenetic regulation may provide insight into disease origins and long-term health. The impact of Mediterranean diet pattern adherence (MDA) on fetal gut microbiome and epigenetic regulation was assessed in 33 pregnant women. Participants completed a validated food frequency questionnaire in each trimester of pregnancy; the alternate Mediterranean diet (aMED) score was applied. Umbilical cord blood, placental tissue, and neonatal meconium were collected from offspring. DNA methylation patterns were probed using the Illumnia EPICarray Methylation Chip in parturients with high versus low MDA. Meconium microbial abundance in the first 24 h after birth was identified using 16s rRNA sequencing and compared among neonates born to mothers with high and low aMED scores. Twenty-one mothers were classified as low MDA and 12 as high MDA. *Pasteurellaceae* and *Bacteroidaceae* trended towards greater abundance in the high-MDA group, as well as other short-chain fatty acid-producing species. Several differentially methylated regions varied between groups and overlapped gene regions including NCK2, SNED1, MTERF4, TNXB, HLA-DPB, BAG6, and LMO3. We identified a beneficial effect of adherence to a Mediterranean diet on fetal in utero development. This highlights the importance of dietary counseling for mothers and can be used as a guide for future studies of meconium and immuno-epigenetic modulation.

## 1. Introduction

The in utero environment has a meaningful impact on the short- and long-term health of a fetus. Maternal nutrition is a well-documented factor that influences the in utero environment and subsequently, optimal child development [1,2,3,4]. One proposed mechanism through which it accomplishes this is by altering the maternal and neonatal gastrointestinal microbiomes [5,6,7]. 

While the intrapartum events and mode of delivery are known to affect the type of bacteria colonizing the neonatal gut, there are recent studies demonstrating that the intrauterine environment is not sterile and that the neonatal gut may be colonized prior to contact with the mother’s birth canal [8]. Several studies have established that antenatal factors directly impact the first-pass meconium of neonates: prenatal maternal diet and exercise, maternal stress, presence of maternal diabetes, use of antibiotics, and the maternal gut microbiome itself all impact the type of microbes present in neonatal meconium [9,10,11,12]. 

The Mediterranean diet (MD) prioritizes the consumption of fruits, vegetables, legumes, nuts, low-processed cereals, olive oil, moderately high levels of fish, and moderate levels of alcohol, in conjunction with low consumption of saturated fat, meats, and dairy products [13,14,15,16]. It has been shown that consumption of plant-based diets is correlated with increased levels of short-chain fatty acids (SCFA) in fecal samples due to increased levels of *Prevotella* and *Firmicutes* [17]. *Prevotella* (a *Bacteroidetes*) and *Lachnospira* (a *Firmicute*) are believed to be able to metabolize complex carbohydrates that are indigestible by hosts through fermentation, thereby increasing SCFAs such as acetate, butyrate, and propionate [18,19,20]. 

Plant-based or fiber-rich diets are also thought to translate to benefits to the fetus in utero as well. For instance, maternal fiber intake and diets rich in fruits and vegetables during pregnancy had the potential to alter the infant’s gut microbiome [21]. In a study performed by Fan et al., higher maternal consumption of fructose, dietary fiber, folic acid, and ascorbic acid was inversely associated with the abundance of *Erysipelatoclostridium* (a *Firmicute*), *Lachnospiraceae* (a *Firmicute*), and *Betaproteobacteria* (a *Proteobacteria*). Chu et al. reported that maternal high-fat diets during gestation and lactation were associated with the depletion of *Bacteroides*, (a SCFA producer) in meconium which was obtained within 24–48 h of delivery [22].

The biological mechanisms that establish associations between maternal diet, the microbiome, and offspring health may occur through neonatal DNA methylation in utero, leaving an environmental imprint on gene regulation for the offspring later in life [23]. DNA methylation may be modified by several environmental factors, including air pollution, stress, diet, and microbial metabolites, such as SCFA [24,25,26]. SCFAs, which are increased in those consuming a fiber-rich diet, such as the Mediterranean-type diet, and also produced by gastrointestinal bacteria, are important molecules in epigenetic regulation [12]. They inhibit histone deacetylase activity, which increases the expression of certain target genes [27]. Additionally, SCFAs influence DNA methylation by modulating the levels of acetyl coenzyme A, and the enzymatic activity of 10–11 translocation methylcytosine deoxygenases [28]. As the maternal diet is known to impact the both neonatal gut microbiome [22] and epigenetic profiles [29], it is reasonable to suspect that associated microbial metabolites may play a direct role in epigenetic regulation. Although the Mediterranean diet is recognized as one of the most effective diets for disease prevention [15,30], current studies have inconsistent findings when examining the impact of maternal adherence to the Mediterranean diet in pregnancy on infant DNA methylation [31,32,33], with varying Cpg sites and neonatal outcomes identified. These studies have not considered the interaction with microbial species.

To our knowledge, there are no studies that have investigated the impact of maternal Mediterranean diet adherence on gut microbiome composition, and subsequently, DNA methylation patterns in infants. In this study, we assessed the associations between maternal Mediterranean diet adherence (MDA) and differential patterns of methylation in the peripheral blood mononuclear cells of neonates. We hypothesize that the maternal diet pattern affects in utero exposure to microbes, and consequently neonatal microbiome in the first 24 h of life. Furthermore, we investigated the associations between maternal diet and neonatal microbiome on neonatal epigenetic programming in neonatal cord blood and the placenta.

## 2. Materials and Methods

This study is a secondary analysis of a prospective cohort study recruited to investigate the effects of maternal dietary patterns on gastrointestinal microbial composition during pregnancy. The protocol has been previously described [34]. Briefly, 41 participants from the four largest ethnic groups in Hawaii (Filipino, Native Hawaiian, Japanese and non-Hispanic White), were recruited in the 1st trimester of pregnancy. Inclusion criteria included age of 18–45 years old, primarily English speaking and English literate, self-identifying as Asian, non-Hispanic White, or Native Hawaiian on intake registration information form, and being in the first trimester of pregnancy (<14 weeks 0 days gestation). After approaching participants, ethnicity was verified via self-reporting as 50% or more of one of the listed ethnicities, or any percentage of Native Hawaiian. Exclusion criteria were plans to move out of the area prior to delivery or to deliver at another hospital than the study institution, multiple gestation, pre-existing diabetes or hypertension, heart disease, chronic renal disease, systemic lupus erythematosus, hypothyroidism, history of bariatric surgery, history of an eating disorder, or inflammatory bowel disease. Women who were currently incarcerated were also excluded from the study. Institutional review board approval was obtained through the Western Institutional Review Board

Participants completed food frequency questionnaires (FFQ) during each trimester, from which dietary patterns were assigned. The Multiethnic Food Frequency Questionnaire (MEC-FFQ), a validated FFQ within our community, was utilized. The MEC-FFQ was developed and validated in a large healthy adult population from 1993–1996 in Hawai‘i and California [35]. The tool is effective in identifying diet patterns associated with mortality and cardiovascular risk [30]. The FFQ includes 182 specific food items uniquely associated with traditional diets such as poi, taro, spam, tofu, salted fish, miso soup, and saimin. Nutritional analysis was performed by the University of Hawaii Cancer Center Nutritional Support Shared Resource, by which an alternate Mediterranean diet score (aMED) was assigned, according to the pattern described by Fung et al. [13], adjusted for energy intake [36]. Components of the diet pattern and scoring algorithm are listed in Table 1. Data output also provided information regarding 54 nutrients from food, energy, macronutrients, and 24 nutrients from supplements.

The aMED is a scale from 1–9, with higher scores representing better adherence. Participants were designated as having high or low adherence to a Mediterranean diet pattern based on being above or below the median aggregate score of all FFQs analyzed, which was a cutoff of a score of 4. No change in scores was noted throughout gestation in the parent study, and thus the average aMED score from all 3 trimesters was used to calculate each aMED score. Those at or above 4 were considered to have high MDA, and those below 4 were considered to have low MDA.

### 2.1. Sample Collection

Offspring of enrolled mothers were sampled at time of delivery. Specimens included neonatal cord blood, a placental specimen, and neonatal meconium, which were collected within the first 24 h after birth. Umbilical cord blood was drawn from the umbilical vein in a sterile manner after clamping of the umbilical cord. It is standard at our institution to allow 30 s of delayed cord clamping, which was performed on all neonates in this cohort. Cord blood was collected in an EDTA tube and centrifuged in order to save peripheral blood mononuclear cells (PBMCs) and plasma separately. Placental specimens were collected after delivery by excising a 1 × 1 inch full thickness portion from each quadrant of the placenta. Membranes were excised from the block and specimens were immediately frozen. Neonatal meconium was collected from the infant diaper, using Copan Eswab. Specimens were also immediately frozen after collection to be processed simultaneously a later date to eliminate batch effects.

Maternal and neonatal characteristics were collected via medical chart review, including maternal age, pre-pregnancy body mass index (BMI), gestational weight gain, pregnancy complications, mode of delivery, birth weight of offspring, and use of antibiotics during pregnancy. Gestational weight gain was characterized as excess or not in excess according to the Institute of Medicine Weight Gain in Pregnancy guidelines, which are based on BMI [37].

### 2.2. Sample Preparation

#### 2.2.1. Illumina Epicarray-Based DNA Methylation Analysis

Neonatal cord blood and placental specimens were subjected to genomic DNA extraction using the Qiagen DNA/RNA Allprep kit (Qiagen, Hilden, Germany). Bisulfite conversion and methylation chip array processing were performed by the University of Hawaii Genomics Core using the MethylationEPIC BeadChip array (Illumina, San Diego, CA, USA), which analyzes more than 850,000 cytosine-guanine dinucleotide (CpG) sites across the genome. Methylation data were acquired using the iScan system (Illumina) as .idat files and processed in R by the EWASTools packages (https://github.com/hhhh5/ewastools, accessed on 2 November 2023) in RStudio (V.4.2.0; www.R-project.org, accessed on 2 November 2023). The manifest file was annotated using IlluminaHumanMethylationEPICanno.ilm10b2.hg19.

#### 2.2.2. 16s rRNA Sequencing

DNA was isolated from meconium samples using the QIAamp Power Fecal Pro DNA Kit. The Ion torrent platform was used for 16s rRNA amplicon sequencing using primers for theV2–V9 regions [26]. Ion Reporter™ Software v5.18.4.0 (ThermoFisher Scientific, Kapolei, HI, USA) was used for assembly, mapping to reference databases in Greengenes v13.5 and MicroSEQ ID v3.0. α-diversity values according to Shannon, Simpson and Chao-1 indices were computed via IonReporter v5.18.4.0.

### 2.3. Data Analysis

The primary objective of this secondary analysis was to evaluate the effect of MDA on neonatal epigenetic programming and its association with neonatal meconium composition. The original study enrolled 41 participants as an exploratory pilot study to characterize the gut and vaginal microbiomes during pregnancy in the Pacific. Baseline demographics of the participants were summarized by mean and standard deviation for continuous variable, frequencies, and percentages for categorical variables. The two-tailed Student’s *t* test, ANOVA or χ^2^ or Fisher’s exact test were used to test the differences of these variables, respectively. Non-parametric tests (Wilcoxon, Kruskal–Wallis) were applied for non-normally distributed data. Beta diversity profiles were analyzed with PCA among each ethnic group during each trimester after the Euclidean distance matrix was developed. The primary outcome measures of correlation of aMED score with alpha diversity score were compared.

An epigenome-wide association study (EWAS) was performed on neonatal cord blood and placenta tissue. After sample processing and image collection as described above, the EWAStools pipeline in R (Version 4.2.0) was used to process, normalize, and analyze global methylation across samples, as well as differential methylation at specific CpG sites and regions. EWAStools is an R package for comprehensive quality control and analysis of DNA methylation microarrays. Quality control metrics performed included those with the standard quality control Array Controls Reporter Version 1.1 Software from Illumina, Dye Bias Correction, and masking of undetected probes at a cutoff of <0.01. Samples were also assessed for SNP outliers for quality control. Leukocyte composition was identified for the cord blood specimens and used as a covariate in the methylation modeling. We removed a total of 22,525 probes with detection *p* values > 0.01 in at least one sample, or a bead count <3 in at least 5% of samples, probes on the X or Y chromosome, or cross hybridizing probes. This yielded 843,393 autosomal probes from 21 samples. The cell composition of each cord blood sample (B cell, CD4T, CD8T, granulocyte, monocyte, natural killer cell) was analyzed using methods described by Houseman et al. [38]. Duplicate samples were verified with SNP genotypes. We used the beta value (β) for our analysis, representing the ratio of the methylated probe intensity and overall intensity (sum of methylated and unmethylated probe intensities). The *p*-value was adjusted to correct for multiple testing using the Benjamini–Hochberg method (false discovery rate, or FDR). Overall global methylation was analyzed using the LimmaEWAS package using beta values, as well as the cpgassoc R package. Differential methylated regions (DMR) were identified using DMRcate. Models included maternal age, obesity, and ethnicity as covariates for the EWAS global methylation and DMRs. All analyses were performed in Rstudio (version 4.2.0).

With regards to meconium microbial composition and diversity, relative abundance of OTUs and assigned species were compared between high and low aMED adherence, with the Mann–Whitney U test used to compare the relative abundance in both groups. Alpha diversity metrics (Shannon, Simpson, and chao) were compared among high vs. low aMED scores, and the PCoA plot using the Euclidean distance matrix used to estimate beta diversity among MDA, obesity, gestational weight gain, and mode of delivery. The EWAS model was run with the *Bacteroides* differential abundance as the predictor variable in order to determine any association with particular neonatal meconium microbial species and intrauterine epigenetic variation.

## 3. Results

From the original cohort of 41 maternal participants, paired neonatal specimens and maternal FFQ dietary information were available for 33 mother–infant pairs, with 28, 24, and 21 mother–infant pairs, respectively, to obtain matched data for neonatal cord blood, placental specimens, and meconium microbial analysis. In the entire cohort of 33 mothers, 21 mothers scored as low adherence and 12 as high adherence. Demographic characteristics are described in Table 2, as well as average macro and micronutrients obtained from the food frequency questionnaire.

### 3.1. Meconium Microbiome Results

Twenty-one meconium samples were analyzed. Average read counts were 290,000; 458 OTUs were characterized at the species level. Alpha diversity was not significantly different as measured by Chao, Simpson, and Shannon Index (Figure 1), among offspring born to mothers with high vs. low aMED scores (as well as maternal gestational weight gain (Appendix A). A beta diversity plot is shown in Figure 2; no significant grouping was noted according to aMED adherence or mode of delivery (as well as maternal obesity or maternal gestational weight gain (Appendix A). Relative abundance was compared at the family, genus, and species level using Mann–Whitney U test. Overall distribution of bacterial abundance at the genus level is depicted in Figure 3. At the family level, there was significantly more abundance of *Pasteurellaceae* in the high aMED group (*p*-value = 0.034), as well as trends for more abundant *Acidaminococcaceae* and *Bacteroidaceae* (Figure 4). At the genus and species level, no significant differences were detected; however, trends were observed of higher *Clostridium lavalense* and *Roseburia intestinalis*. The most abundant species within all samples was *Faecalibacterium prausnitizii*. No differences were detected at the species, genus, or family level among mothers who gave birth vaginally (*n* = 17) vs. by cesarean section (*n* = 4).

### 3.2. Methylation Analysis

Twenty-nine neonatal cord blood samples and 24 placenta samples were processed via the Illumnia EPic Array Methylation Chip and analyzed (separately) with exposures of high versus low aMED adherence. All neonatal cord blood samples passed quality control; none needed to be excluded from final analysis. Four samples were excluded from the placental tissue analysis, leaving a remainder of 20 participants for EWAS analysis of placenta tissue. Overall global methylation did not differ significantly between the two groups in either sample type. No specific CPG sites met the threshold of FDR <0.01 in cord blood. When performing DMR analysis, there were specific differentially methylated regions identified among high versus low MDA groups, listed in Table 3. The final model is adjusted for obesity, ethnicity, and parity.

Meconium bacterial abundance was available for 21 of the neonates from the cohort, and neonatal cord blood and placenta were available for 18 of the neonates. To look at the interaction between bacterial abundance as a predictor of methylation, we investigated DMRs in the subset of participants (*n* = 18) with meconium microbiome data available. Participants with low vs. high abundance of *Bacteroides*, as surrogate marker for higher levels of SCFA production, were analyzed to look for differential methylation patterns. There was one DMR that differed significantly in cord blood among neonates with higher amounts of *Bacteroidaceae* vs. low read counts, also listed in Table 3.

## 4. Discussion

Adherence to a Mediterranean diet is associated with several health benefits outside of and during pregnancy [39,40,41]. Recommending a Mediterranean diet during pregnancy may improve both maternal and neonatal health outcomes [42,43,44]. This study aimed to characterize potential beneficial biologic features of MDA in utero, specifically of the low biomass microbes that a fetus is exposed to during gestation and the impact of microbial metabolites on epigenetic regulation, which may portend lifelong translational effects.

Our findings of MDA suggest beneficial effects of neonatal meconium composition, including increases in *Ruminococcaceae* (a *Firmicute*), *Acidaminococcaceae* (a *Firmicute*), and *Bacteroidaceae* (a *Bacteroidetes*) abundance. At the species level, *Roseburia Intestinalis* and *Clostridium lavalense* trended towards a statistically significant difference among high vs. low aMED consumption groups. *Faecalibacterium prausnitizii* was the most abundant species identified in both groups. This organism is recognized as one of the most abundant species found in the healthy human microbiota [18]. *Faecalibacterium prausnitzii* is well-known for its anti-inflammatory effects, found in both in vitro and in vivo studies [45]. It produces a microbial anti-inflammatory molecule (MAM) that protects against inflammatory bowel disease, restores mucosal intestinal barrier integrity, and consumes acetate from neighboring bacterial to produce butyrate [46,47]. These molecules help with mucin glycosylation to aid in intestinal mucosal integrity. Although no statistical difference was found between low and high MDA, there is an observed trend for higher counts of *Faecalibacterium prausnitzii* in high-MDA mothers (Figure 4). Further research in larger cohorts would be useful to investigate further the link between maternal MDA and increase of *Faecalibacterium prausnitzii*, and subsequent presence of anti-inflammatory molecules and SCFA.

The authors acknowledge that mode of delivery is thought to be a strong contributor to meconium microbial composition [48]. Previous literature has documented differences in bacterial species present in the meconium of infants who were born vaginally versus by cesarean, but results are mixed [49,50,51]. For example, Shi et al. found that the most abundant phyla in vaginally delivered infants were *Firmicutes* and *Deinococcus-Thermus*, while *Actinobacteria* were most abundant in C-section-delivered infants [49], while Weng et al. found that neonates born by cesarean had meconium that was primarily dominated by *Enterococci* [50]. Contrarily, Martin et al. found that infants born by C-section had a lower prevalence of *Enterococci*, *Bacteroides*, and *Clostridium* species in their meconium [51]. With regards to overall diversity, our study identified significantly higher Shannon and Simpson diversity in infants born by cesarean compared to those born vaginally. This is in contrast to Shi’s study, which found that the diversity and richness of the neonatal meconium was higher in vaginally delivered infants compared to cesarean-delivered infants [49]. The largest study to date was likely performed by Tapiainen et al. [11]. They noted no difference in microbial composition between vaginal versus cesarean birth when comparing meconium of 218 infants after 24 h of life. They concluded that colonization of the gut microbiome likely occurs prior to delivery; thus, meconium bacterial composition may be independent of delivery mode. Ultimately, current studies lacked power to detect which factor—mode of delivery, obesity, maternal diet—has the strongest effect of meconium microbial composition, and further research is needed.

Several studies have looked at maternal diet and epigenetic programming in the offspring also with inconsistent findings [52,53,54]. Human studies assessing maternal diet and neonatal epigenetic changes have investigated the role of polyunsaturated fats [52,55], folate [2], and low-glycemic index diets [54], with varying results. Recently, Küpers et al. performed an EWAS meta-analysis and reported associations between maternal MDA during pregnancy and increased offspring cord blood methylation of one CpG, cg23757341. This CpG site can be mapped to the transcription start site of the WNT5B gene, which has been associated with adipogenesis, insulin secretion, and type 2 diabetes [56]. Gonzalez-Nahm et al. found an association between low maternal MDA and higher odds of female infant hypomethylation at the MEG3-IG region, believed to be the upstream regulator of the MEG3 DMA in association with type 2 diabetes [31].

Our study also found beneficial differences in neonatal cord blood methylation in infants born to mothers with high MDA, including overlapping regions with the NCK2, SNED1, MTERF4 and MSH5, and HLA-DPB1 genes. Some metabolically relevant candidate areas include the differential methylation across CpG sites in the region of the NCK2 gene. Animal studies reported that NCK deficiency was associated with increased adiposity, impaired adipocyte function, glucose intolerance, insulin resistance, and hepatic steatosis [57]. An epigenome-wide analysis found that DNA-methylation signatures in the SNED1 gene were significantly associated with lipid and glucose metabolism, diabetes mellitus, body size, and body composition in European children [58].

Our analysis also identified one statistically significant variation in placenta methylation: LIM domain only 3 (LMO3). LMO3 is expressed in adipocytes and is thought to regulate genes that promote adipose tissue functionality in obesity [59,60].

While investigation of epigenetic-microbial interactions is of growing interest in cancer, obesity, and neurodevelopmental research [28,61,62], this is an uncharted question in the field of developmental programming. Bacterial populations in a host can affect epigenetic regulation and correlate with clinical outcomes: gut microbial dysbiosis leads to inflammation, decrease in SCFA, and increased risk of inflammatory bowel disease IBD) [63]. A study evaluating self-esteem in patients with type 2 diabetes noted similar gut microbiome profiles, inflammatory profiles, and regions of differential methylation in participants with low self-esteem [64]. When we looked at associations with the level of *Bacteroidaceae* abundance and epigenetic changes, we found one particular DMR that varied between high vs. low abundance. This region on chromosome 6 is not far from another region identified to be differentially regulated in association with MDA and that overlaps with the BAG6 gene. This gene was previously identified in a large genome-wide association study investigating the association between neuropsychiatric symptoms and irritable bowel syndrome [65]. Overall, this region may be an interesting candidate for future research.

The authors acknowledge the limitations of the small sample size and varied mother–infant matched pairs among tissue types in this study. The topic of MDA and offspring epigenetic programming is broad and heterogeneous, and these explorations are preliminary, requiring larger cohorts. Other limitations of this study are that we did not directly measure amounts of SCFA in meconium, and thus used bacteria known as SCFA producers as a surrogate for the presence of SCFA. While fetal sex chromosomes were used for quality control metrics, we did not use this covariate in the EWAS model. Finally, one may wonder why we did not investigate the impact of the maternal microbiome on epigenetic regulation of the neonate. Without knowing how in utero microbial signatures are populated (from maternal gut, vaginal, oral cavity), and in what part of gestation, we felt that it was more powerful to use the signatures of organisms that were already present, in close proximity, to the offspring tissue being studied. Thus, we believe neonatal gut microbes within 24 h, presumably characteristic of in utero microbial populations, could be more representative and influential on the epigenome.

There are limited studies investigating bacterial abundance on epigenome-wide methylation [61,65,66], especially in neonatal cohorts, which makes this study unique. With increased bioinformatic and data science techniques, there is greater ability to identify low-biomass bacterial signatures through next-generation sequencing. We aimed to shed light on microbial–host communication during the in utero timeframe. Immunoepigenetic crosstalk between dietary molecules, intestinal microbes, and host genomes are just beginning to be understood in adults, and fetal life may be a time in which this communication is primed. For instance, researchers used a cell culture model to assess epigenome–microbiome crosstalk in preterm infants at risk of necrotizing enterocolitis [67]. They exposed enterocytes to bacteria known to be both beneficial and pathogenic to neonates and assessed the effects on epigenetic regulation. They found more than 200 regions of differential DNA modification related to the exposures.

## 5. Conclusions

In summary, we identified a beneficial effect on fetal development from maternal Mediterranean diet adherence, including beneficial microbial signatures and differential epigenetic regulation. The outcome of these associations warrants further investigation, such as of interactions between various intestinal bacteria, and as well as functional metagenomics of the organisms. Our findings highlight the importance of adopting a Mediterranean-type diet, especially during pregnancy, and should be an impetus to support pregnant persons with the ability to access fresh, whole foods in line with this diet pattern. Furthermore, public health initiatives that focus on food-as-medicine interventions should consider the importance of a plant-based, Mediterranean-type diet.

## Figures and Tables

**Figure 1 nutrients-16-00047-f001:**
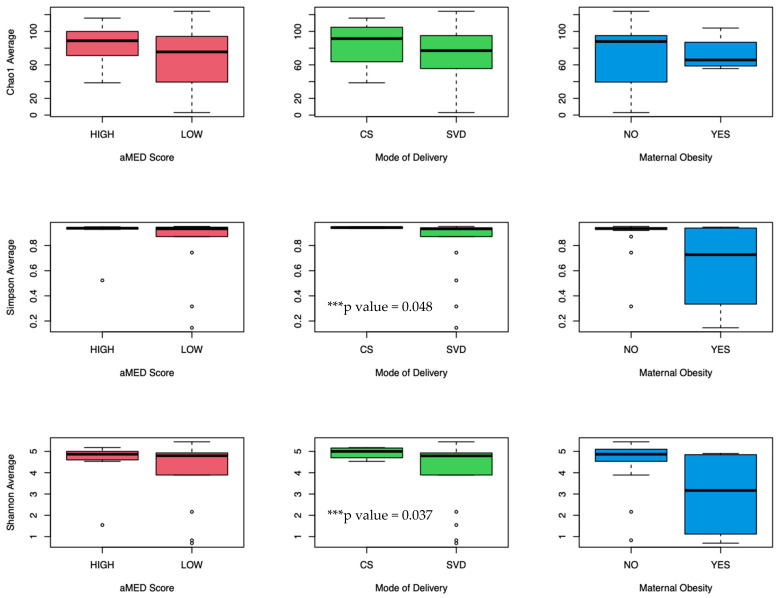
Alpha diversity metrics (Chao1, Simpson, and Shannon Indices) of neonatal meconium microbiome among high vs. low aMED scores, mode of delivery (CS = cesarean section, SVD = spontaneous vaginal delivery), and presence of maternal obesity (defined as maternal pre-pregnancy BMI > 30 mg/kg^2^). Metrics were compared via Mann–Whitney U test, statistically significant comparisons = ***, including Simpson and Shannon indices according to mode of delivery (0.943 (CS) vs. 0.815 (SVD), *p* = 0.048; 4.93 (CS) vs. 4.00 (SVD) *p* = 0.037, respectively).

**Figure 2 nutrients-16-00047-f002:**
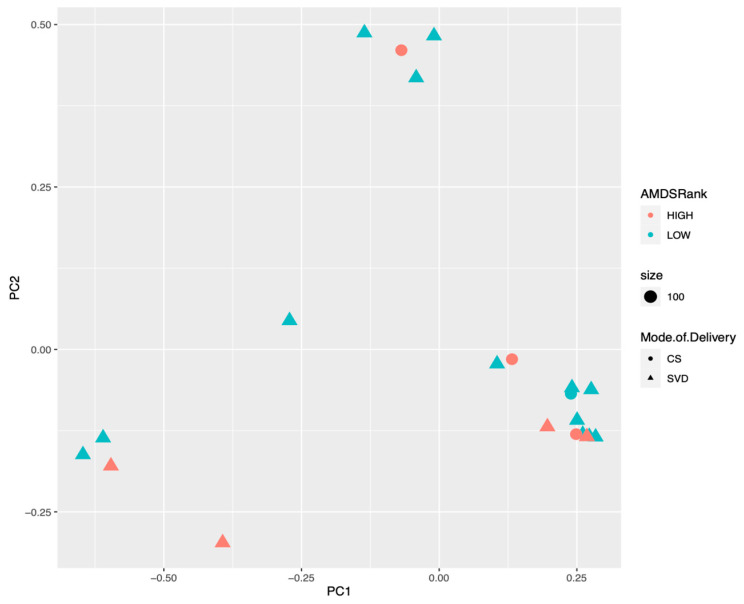
Beta diversity of neonatal meconium samples compared by maternal aMED score and mode of delivery.

**Figure 3 nutrients-16-00047-f003:**
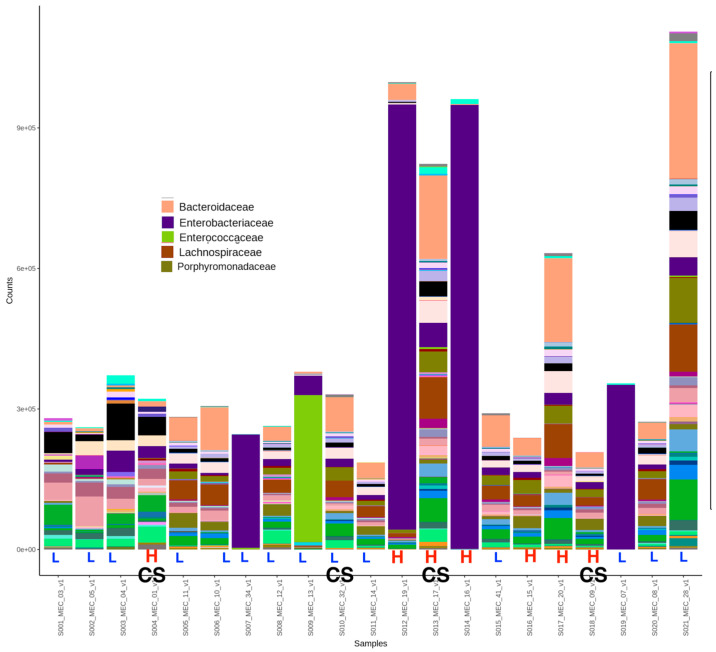
Distribution of OTUs at genus level by sample (each column). Legend depicts the most abundant organisms at the genus level. Mode of delivery and aMED adherence is demonstrated at the bottom of the panel (mode of delivery—CS = cesarean section, all other samples delivered vaginally). aMED scores are classified as high (H) or low (L).

**Figure 4 nutrients-16-00047-f004:**
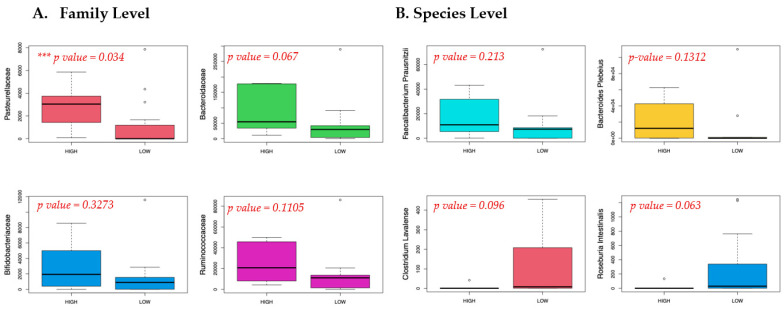
Relative abundance at the family level (**A**) and species level (**B**), depicting some of the highest abundance or largest differences among low vs. high MDA participants.

**Table 1 nutrients-16-00047-t001:** Scoring components for alternate Mediterranean diet (aMED) score adapted from Fung et al. (2005) [13].

Food Group	Foods Included	Criteria for 1 Point
Vegetables	All vegetables except potatoes	Greater than median intake (servings/d)
Legumes	Tofu, string beans, peas, beans	Greater than median intake (servings/d)
Fruit	All fruit and juices	Greater than median intake (servings/d)
Nuts	Nuts, peanut putter	Greater than median intake (servings/d)
Whole Grains	Whole-grain cereals, cooked cereals, crackers, dark breads, brown rice, other grains, wheat germ, bran, popcorn	Greater than median intake (servings/d)
Red and Processed Meats	Hot dogs, deli meat, bacon, hamburger, beef	Less than median intake (servings/d)
Fish	Fish and shrimp, breaded fish	Greater than median intake (servings/d)
Ratio of Monounsaturated to Saturated Fat	-	Greater than median intake (servings/d)
Ethanol	Wine, beer, light beer, liquor	<25 g/d

**Table 2 nutrients-16-00047-t002:** Baseline demographics and dietary measures of participants. Excess gestational weight gain was defined as more than recommended amount of weight as recommended by Institute of Medicine according to body mass index [37] (SD = standard deviation). All nutritional components are averaged from all 3 trimesters.

	Low aMED Adherence (*n* = 21)	High aMED Adherence (*n* = 12)	*p*-Value
**Maternal Age** Median (SE)	28 (5.2)	33 (5.5)	0.123
**Ethnicity**			0.078
Non-Hispanic White	6	3	
Filipino	7	0	
Native Hawaiian	3	5	
Japanese	5	4	
**Parity**			0.452
Nulliparous	12/21	7/12	
Primiparous	8/21	3/12	
Multiparous	1/21	2/12	
**Pregnancy Complications**			0.064
Gestational Diabetes	1/21	2/12	
Preeclampsia	5/21	2/12	
Preterm Labor	1/21	0/12	
**Maternal Obesity** ^1^	29.41%	25%	0.717
**Gestational Weight Gain (Mean, +/− SD)**	28.04 (9.89)	26.83 (12.03)	0.761
**Excess Gestational Weight Gain**	5	4	0.503
**Mode of Delivery**			
Vaginal (Spontaneous or Operative)	17/21	7/12	0.071
Cesarean Delivery	4/21	5/12	
**Neonatal Birth Weight (g)**	3206.05 g (+/− 468.35)	3412.67 g (+/− 495.57)	0.251
**Gestational Age at Delivery (Median in weeks)**	39 weeks	39 weeks	1.000
**Total Kilocalories/day (mean, [+/− SD])**	2297.9 (1267.18)	1638.4 (612.21)	0.238
**Macronutrients**			
% Energy from Carbohydrates	46.33%	51.20%	0.139
% Energy from Total Fat	36.78%	34.00%	**0.015**
% Energy from Protein	16.88%	14.78%	0.415
**Micronutrients** Mean (SD)			
Fiber (g)	14.10 (7.11)	31.18 (20.04)	**0.003**
Vitamin D (International Units)	126.00 (67.92)	159.00 (105.46)	0.275
Vitamin B12 (mcg)	5.12 (2.44)	7.09 (4.72)	0.120
Monosaturated Fatty Acids (g)	27.5 (11.75)	32.16 (13.43)	0.312
Saturated Fatty Acids (g)	24.28 (9.78)	27.112 (10.12)	0.440
Monounsaturated:Saturated Fatty Acid Ratio	1.13 (0.16)	1.17 (0.17)	0.508
Polyunsaturated Fatty Acids (g)	12.44 (5.27)	18.93 (10.3)	**0.025**

^1^ Obesity defined as body mass index > 30 mg/kg^2^, g = grams, mcg = micrograms.

**Table 3 nutrients-16-00047-t003:** Differentially methylated regions of neonatal cord blood and overlapping genes as mapped on NCBI.

Tissue	Chromosome	Base Pair Region	Overlapping Gene Symbol	FDR
Cord Blood— aMED Model	Chr1	67600546–67600963		5.68 × 10^−7^
	Chr1	108023248–108023486		1.27 × 10^−8^
	Chr2	105853199–105853526	NCK2, ENSG00000235522	2.53 × 10^−7^
	Chr2	241076415–241076601	SNED1, MTERF4	4.63 × 10^−8^
	Chr3	133502622–133502917		1.73 × 10^−7^
	Chr4	74847709–74848016		6.89 × 10^−9^
	Chr6	31744523–31744628	MSH5, MSH5-SAPCD1	1.14 × 10^−4^
	Chr6	32063873–32064146	TNXB	7.17 × 10^−5^
	Chr6	33084419–33085063	HLA-DPB1	4.05 × 10^−15^
	Chr7	30196738–30197130		1.04 × 10^−10^
Cord Blood— Bacteroides	Chr6	31650734–31651158	BAG6	1.03 × 10^−11^
Placenta	Chr12	16758934–16759391	MGST1, LMO3	2.64 × 10^−9^

## Data Availability

Following the acceptance of our manuscript for publication, these datasets will be deposited into appropriate databases including the NCBI Gene Expression Omnibus (GEO) database, the NCBI Short Read Archives (SRA), MicrobiomeDB, and other relevant databases and made freely available to investigators at academic institutions worldwide.

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
