# Peer review of "Impact of Maternal Mediterranean-Type Diet Adherence on Microbiota Composition and Epigenetic Programming of Offspring"

_nutrients, 2023, doi:10.3390/nu16010047_

Round 1

Reviewer 1 Report

Comments and Suggestions for Authors

I would like to show appreciation to the authors for their contribution to an important scientific area of research. Maternal diet during the pregnancy period is crucial for their independent current and future health, as well as the immediate and long-term health of their offspring. Although I commend the authors for their excellent research design and writing, I have several minor concerns that warrant addressing prior to consideration for publication.

In the abstract, the authors need to distinguish which outcome is derived from what biological specimen to tie the results in more adequately. Further, should tie together that epigenetics are related to the outcomes, specifically DNA methylation. Without this the abstract is lacking some critical detail.

Throughout the entirety of the manuscript, there is a lack of supporting references for in-text statements that require corroboration. I also find the references cited do not always match the statement made within the body of the manuscript. Collectively, a more thorough review of the literature and ensuring proper citation is necessary to alleviate this concern.

Minor consideration as well is the consistent use of terminology and abbreviations/acronyms. For example, intrapartum, antenatal, prenatal for terminology, and MD/aMED could be streamlined.

The Introduction needs a paragraph that specifically discusses the impact of maternal diet on maternal and infant microbiota. This is hinted on, but not fully discussed. I also find the Introduction lacking key detail that truly shows the impact of the work being performed and justify the need for further research on this topic.

The Methods should list critical inclusion and exclusion criteria, even though the authors referenced the protocol manuscript published.

Need to also provide information on the sample proposed in the protocol paper compared to those analyzed in the manuscript. Later, I see where you justify this, but need to speak on how you came to this sample size.

For the FFQ, where all trimesters are listed, you should put the Mean+-SD for gestational age (GA) they completed the FFQ to ensure it aligns with the exact trimester included. Also, did the authors average the FFQ across pregnancy and include in analysis, or individual trimester? Was the FFQ results similar across pregnancy?

Additionally, for the methods, was gestational weight gain (GWG) as a continuous and categorical variable assessed? The authors need to include this as evidence has suggested that maternal body mass index (BMI) and GWG influence maternal and fetal microbiota, independent of diet intake. As such, GWG needs to be discussed, the results presented in the table, as well categorical GWG included as a covariate in analyses performed.

Kilocalorie per day should also be explored for potential outliers in the data. As the aMED is a composite score and is bound by a restrictive range, the data used may not be physiologically relevant. For instance, a woman with overweight has an energy requirement of 2500 kcal/day, but their FFQ resulted in their habitual dietary intake being 6500 kcal/day. This would not be reflected in the aMED and this data should most likely be excluded as an outlier and as non-habitual energy intake. Alternatively, you could include an analysis that excluded outliers and found that this analysis yielded different or similar results, and include as a Supplement. The Redman laboratory at Pennington Biomedical Research Center has published some of these methodologies and I suggest including their methods in this analysis for diet intake. However, this information is needed to justify the results.

The Discussion is too long and is difficult to interpret. Acknowledge that there is a good amount of data to discuss, but summarizing findings more eloquently based on similar results would harmonize the Discussion. Plus, some of the points made in the Discussion would alleviate some of my points made on the Introduction if revised and moved around. Suggest taking some of the Discussion points and adding them to Introduction section.

The Conclusion, while logical, lacks passion in the future directions. Food Insecurity, Food as Medicine Initiatives, Precision Nutrition Health, as examples, are all current buzzwords and booming areas of research. Therefore, the authors should include details like this in their conclusion to highlight that there is a need for further dietary interventions and public policies may be taking shape to help alleviate some of these concerns or challenges with improving maternal diet to influence intergenerational health.

Author Response

Thank you very much for your valuable feedback. Please see the attachment for our responses.

Reviewer 2 Report

Comments and Suggestions for Authors

In this manuscript, the authors., studied the possible association between maternal Mediterranean Diet adherence (MDA) and differential patterns of methylation in the peripheral blood mononuclear cells of neonates and neonatal gut microbiome composition. The focus of work is very interesting. The manuscript is well organized and well written.  

The authors describe the limitations of this work, they should add the need to analyze the data also taking into account the sex of the newborn.

Author Response

(The authors gave the same response as above.)

Reviewer 3 Report

Comments and Suggestions for Authors

To the authors

BRIEF SUMMARY

This study investigated the impact of maternal Mediterranean diet adherence on microbiota
composition and epigenetic programming of offspring.
In mothers with high MD adherence, pasteurellaceae and bacteroidaceae sp, along with other short-chain fatty acid-producing species, were isolated. Several differentially methylated regions were identified. The investigators concluded that adherence to a Mediterranean Diet had a beneficial effect on fetal development.

Overall, a good effort was made by the authors. However, there are issues that require clarification in order to facilitate study transparency. More details are needed in the methods section. In the discussion, references must be added appropriately to support the statements.

Please refer to the comments to the authors.

Comments

Introduction

-Line 37 Add ref after the statement ‘Several studies have established that antenatal

 factors directly impact the first-pass meconium of neonates.’

-Line 42-44 The Mediterranean Diet (MD) prioritizes the consumption of fruits, vegetables,

legumes, nuts, low-processed cereals, moderately high levels of fish, and moderate levels of

 alcohol, in conjunction with low consumption of saturated fat, meats, and dairy product’

An important fundamental characteristic of the Mediterranean dietary pattern is the high intake of virgin olive oil, the main source of monounsaturated fat.

Please add olive oil.

Refer to these articles by

1. Simopoulos AP. The Mediterranean diets: What is so special about the diet of Greece? The scientific evidence. J Nutr. 2001 Nov;131(11 Suppl):3065S-73S. doi: 10.1093/jn/131.11.3065S. PMID: 11694649.

2. Trichopoulou et al., N Engl J Med 2003;348:2599-608

-Line 51-53 add ref after this statement ‘The biological mechanisms that establish associations between maternal diet, the microbiome, and offspring health may also occur through neonatal DNA methylation in- utero, leaving an environmental imprint on gene regulation for the offspring later in life.’

-Line 55-57 add refSCFAs, which are increased in those consuming a Mediterranean Diet and also produced by gastrointestinal bacteria, are important molecules in epigenetic regulation.’

-Line 57-58 add ref ‘They inhibit histone deacetylase activity, which increases the expression of certain target genes.’

-line 63-66, ‘Although the Mediterranean Diet is recognized as one of the most effective diets for disease prevention, current studies have inconsistent findings when examining the impact of maternal adherence to the Mediterranean Diet in pregnancy on infant DNA methylation [18-21], and have not considered the interaction with microbial species’

Elaborate, describe in brief what these studies report.

Methods

Line 78, describe in brief the parent study

-Line 90 ………which alternate Mediterranean Diet Score (aMED) was assigned, according to the pattern described by Fung et al [25].

Describe in detail the alternate Med diet score. Which food groups are included and how the score is derived, as well as cut-off levels considered high vs low adherence

-Was intake of olive oil assessed and included? This is not clear from the manuscript.

If not, this should be mentioned in the methods section and I suggest that the title is changed to a Mediterranean-type diet, given that it is olive oil the main constituent of the Mediterranean diet which differentiates  the Med diet pattern from other healthy diets.

-Line 93, describe the sample (XX mothers were enrolled from ………, XX offspring were assessed at the time of delivery.

-Line 103,  specimens were frozen and stored?????

-Lines 129-134 ‘The  aMED is a scale from 1-10, with higher scores ………. “High” MDA, and those below 6 were considered to have low MDA’

This statement belongs in the methods section not in data analysis.

Results

Table 1- Do you have data on % monounsaturated, polyunsaturated, saturated fats as well as the ratio of monounsaturated to saturated, and ratio of omega 6 to omega 3

-Line 252 At the family level, there was significantly more abundance of Pasteurellaceae in the high AMED group, as well as trends for more abundant Acidaminococcaceae and Bacteroi daceae’ Mention significant p-values

-Line 253, typo error ‘AMED’, perhaps aMED?

-Line 266 ‘4 samples’. Rewrite as Four (4) samples …….

Discussion

-Line 282, ref 24 refers to the study by Kolonel et al. , not a Med diet study. Please revise

-Line 282, check references 28 and 29 if they are appropriate and support your statement.

-Line 288 add references ‘Few studies have looked at the neonatal gut microbiome within the first 24 hours after birth and its association with maternal dietary patterns.’

-Line 291-293, add ref ‘Another study looked at maternal fiber intake and concluded that maternal diets rich in fruits and vegetables during pregnancy had the potential to alter the infant's gut microbiome’

-Line 314, add ref “The authors acknowledge that mode of delivery is thought to be a strong contributor to meconium microbial composition. ‘

-Line 315-317,  add ref ‘Previous literature has documented differences in bacterial species present in infants’ meconium who were born vaginally versus by C-section, but results are mixed.’

-Line 331 –‘aren’t powered’, rephrase. Perhaps ‘lacked power’

-Line 334 add refs, Several studies have looked at maternal diet and epigenetic programming in the off-spring also with inconsistent findings.’

-Line 335 ‘ Human studies looking at’. Better to say Human studies assessing or investigating

-Line 342-344, add ref ‘Gonzalez-Nahm et al. used a gene-specific approach to investigate the relationship between methylation of specific targeted genes and maternal MDA.’

-Line 358-360, add ref, ‘…and in a mouse study, lack of MTERF4 in brown adipose tissue (BAT) resulted in reduced respiratory capacity, and thus a dysregulated thermogenic response when exposed to cold temperatures.’

-Line 364-365 add ref, ‘LMO3 is expressed in adipocytes and is thought to regulate genes that promote adipose tissue functionality in obesity.’

-Line 369-372, add ref ‘A study involving mothers with pregestational diabetes mellitus (PDM) found that mRNA expression of LMO3 was increased in human umbilical cord Wharton’s jelly mesenchymal stem cells from normoglycemic pregnancies (NGP), but not in PDM pregnancies, under hyperglycemic conditions.’

-Line 407-408. Add ref to studies ‘There are limited studies investigating bacterial abundance on epigenome wide methylation, which makes this study unique.’

-Line  413-415 add ref, ‘For instance, researchers used a cell culture model to assess Epigenome-Microbiome crosstalk in pre-term infants at risk of necrotizing enterocolitis.’

Conclusion

-Line 426-428 ‘Our findings highlight the importance of dietary counseling for mothers and can be used as a guide to look at future studies of meconium and immuno-epigenetic modulation’

This study suggests the importance of adopting a Med-type diet, especially during pregnancy.

References

Reference 29, authors’ names must be revised. B, H.A.W., et al., Mediterranean-style diet in pregnant women with metabolic risk factors (ESTEEM): A pragmatic multicentre randomised trial. PLoS Med, 2019. 16(7): p. e1002857

Comments on the Quality of English Language

A couple of minor errors were detected. Please refer to my comments to the authors.

Author Response

(The authors gave the same response as above.)
